# Ciliary Hedgehog signaling regulates cell survival to build the facial midline

**Shaun R Abrams[1,2], Jeremy F Reiter[1,3]***

[1]Department of Biochemistry and Biophysics, Cardiovascular Research Institute, San Francisco, United States; [2]Oral and Craniofacial Sciences Program, School of Dentistry, San Francisco, United States; [3]Chan Zuckerberg Biohub, San Francisco, United States

**Abstract** Craniofacial defects are among the most common phenotypes caused by ciliopathies, yet the developmental and molecular etiology of these defects is poorly understood. We investigated multiple mouse models of human ciliopathies (including *Tctn2, Cc2d2a,* and *Tmem231* mutants) and discovered that each displays hypotelorism, a narrowing of the midface. As early in development as the end of gastrulation, *Tctn2* mutants displayed reduced activation of the Hedgehog (HH) pathway in the prechordal plate, the head organizer. This prechordal plate defect preceded a reduction of HH pathway activation and *Shh* expression in the adjacent neurectoderm. Concomitant with the reduction of HH pathway activity, *Tctn2* mutants exhibited increased cell death in the neurectoderm and facial ectoderm, culminating in a collapse of the facial midline. Enhancing HH signaling by decreasing the gene dosage of a negative regulator of the pathway, *Ptch1*, decreased cell death and rescued the midface defect in both *Tctn2* and *Cc2d2a* mutants. These results reveal that ciliary HH signaling mediates communication between the prechordal plate and the neurectoderm to provide cellular survival cues essential for development of the facial midline.

## Introduction

Primary cilia are microtubule-based organelles present on diverse vertebrate cell types and critical for development. Primary cilia function as specialized cellular signaling organelles that coordinate multiple signaling pathways, including the Hedgehog (HH) pathway (*Zaghloul and Brugmann, 2011*). Defects in the structure or signaling functions of cilia cause a group of human syndromes, collectively referred to as ciliopathies, which can manifest in diverse phenotypes including cystic kidneys, retinal degeneration, cognitive impairment, respiratory defects, left-right patterning defects, polydactyly, and skeletal defects (*Baker and Beales, 2009*; *Hildebrandt et al., 2011*; *Tobin and Beales, 2009*). In addition to these phenotypes, craniofacial defects including cleft lip/palate, high-arched palate, jaw disorders, midface dysplasia, craniosynostosis, tongue abnormalities, abnormal dentition, and tooth number and exencephaly are observed in approximately one-third of individuals with ciliopathies (*Brugmann et al., 2010b*; *Zaghloul and Brugmann, 2011*). The molecular and developmental etiology of these craniofacial abnormalities remains poorly understood.

HH signaling is intimately involved in forebrain and midface development (*Hu and Helms, 1999*; *Rubenstein and Beachy, 1998*). In humans, inherited mutations that compromise pathway activity impair forebrain development and cause hypotelorism (*Fuccillo et al., 2006*; *Hu and Helms, 1999*; *Hu and Marcucio, 2009*; *Marcucio et al., 2005*; *Muenke and Beachy, 2000*; *Young et al., 2010*). For example, mutations in *SHH* lead to holoprosencephaly (*Chiang et al., 1996*; *Cohen and Shiota, 2002*). Meckel syndrome (MKS), a severe ciliopathy, is also characterized by holoprosencephaly and hypotelorism (*Chih et al., 2011*; *Dowdle et al., 2011*; *Garcia-Gonzalo et al., 2011*). MKS-associated genes encode proteins that form a complex that comprises part of the transition zone, a region of the ciliary base that controls ciliogenesis and ciliary membrane protein composition in a tissue-specific

***For correspondence:**
Jeremy.Reiter@ucsf.edu

manner (*Chih et al., 2011*; *Dowdle et al., 2011*; *Garcia-Gonzalo et al., 2011*; *Roberson et al., 2015*). Disruption of this transition zone complex results in the impaired ciliary localization of several membrane-associated signaling proteins including smoothened (SMO), adenylyl cyclase 3 (ADCY3), polycystin 2 (PKD2), and ARL13B (*Chih et al., 2011*; *Garcia-Gonzalo et al., 2011*; *Roberson et al., 2015*).

We investigated the molecular underpinnings of forebrain and midface defects in ciliopathies utilizing multiple mouse mutants affecting the transition zone. The mutants exhibited forebrain and midface defects by E9.5, which persisted throughout development. In these mutants, the prechordal plate, an organizer of anterior head development, displayed defects in HH pathway activation at E8.0. These early prechordal plate defects attenuated *Shh* expression in the adjacent ventral forebrain. Decreased HH signaling increased apoptosis in the ventral neurectoderm and facial ectoderm. Surprisingly, reducing *Ptch1* gene dosage rescued the apoptosis and its corresponding midface defect. Thus, investigating the function of the transition zone has revealed a key role of prechordal plate-activated HH signaling in forebrain and midface cell survival. Moreover, our genetic results reveal that inhibition of PTCH1 can prevent ciliopathy-associated midface defects. Based on these mouse genetic models, we propose that the etiology of hypotelorism in human ciliopathies is a failure of the prechordal plate to induce SHH expression in the overlying ventral neuroectoderm.

## Results
### The ciliary MKS transition zone complex is essential for midline facial development

Individuals affected by developmental ciliopathies, such as Meckel, orofaciodigital, and Joubert syndromes, often display craniofacial phenotypes including holoprosencephaly and hypotelorism (*Baker and Beales, 2009*; *Dowdle et al., 2011*; *Garcia-Gonzalo et al., 2011*). To explore the etiology of these craniofacial defects, we examined the craniofacial development in *Tctn2* mouse mutants (*Garcia-Gonzalo et al., 2011*; *Shaheen et al., 2011*). TCTN2 is a component of the MKS transition zone complex critical for ciliary localization of several ciliary membrane proteins, including SMO, a key ciliary mediator of HH signaling (*Chih et al., 2011*; *Corbit et al., 2005*; *Garcia-Gonzalo et al., 2011*). Mutations in human *TCTN2* cause Meckel and Joubert syndromes (*Huppke et al., 2015*; *Sang et al., 2011*; *Shaheen et al., 2011*).

*Tctn2*[+/-] embryos were phenotypically indistinguishable from *Tctn2*[+/+] embryos (*Figure 1—figure supplement 1*). In contrast, embryonic day (E) 10.5 *Tctn2*[-/-] embryos exhibited an approximately 50 % decrease in infranasal distance (the distance between the nasal pits, the nostril anlage) (*Figure 1A*). One day later in gestation (E11.5), *Tctn2*[-/-] embryos also exhibited midfacial narrowing, including hypoplasia of the frontonasal prominence and fusion of the two maxillary prominences at the midline (*Figure 1A*). Thus, TCTN2 is essential for development of the facial midline.

To assess whether this narrowing of the facial midline is specific to TCTN2, we examined the possible involvement of two other components of the MKS complex, TMEM231 and CC2D2A, in craniofacial development. Human *CC2D2A* mutations cause Meckel and Joubert syndromes, and *TMEM231* mutations cause Meckel, Joubert, and Orofaciodigital syndromes (*Noor et al., 2008*; *Roberson et al., 2015*; *Shaheen et al., 2013*; *Srour et al., 2012*; *Tallila et al., 2008*). Similar to *Tctn2* mutants, both E10.5 *Cc2d2a* and *Tmem231* mutant embryos exhibited decreased infranasal distance (*Figure 1B and C*, respectively). The similarity of the midline hypoplasia in all three transition zone mutants suggested a common etiology.

We also examined the involvement of a fourth member of the MKS complex, TMEM67, in craniofacial development. Human mutations in *TMEM67* also cause Meckel and Joubert syndromes (*Otto et al., 2009*; *Smith et al., 2006*). Mutation of mouse *Tmem67* causes phenotypes that are less severe than *Tctn2*, *Tmem231*, or *Cc2d2a* (*Garcia-Gonzalo et al., 2011*). The mild phenotype of *Tmem67* mutants may be attributable to its dispensability for ciliary accumulation of HH pathway activator SMO (*Garcia-Gonzalo et al., 2011*). Unlike *Tctn2*, *Tmem231*, and *Cc2d2a* mutants, *Tmem67* mutants did not exhibit altered infranasal distance (*Figure 1D*). Thus, some, but not all, MKS components are critical for early facial midline development.

Given the central role of the neural crest in craniofacial development as the main source of craniofacial mesenchyme (*Santagati and Rijli, 2003*), we tested whether transition zone disruption in neural

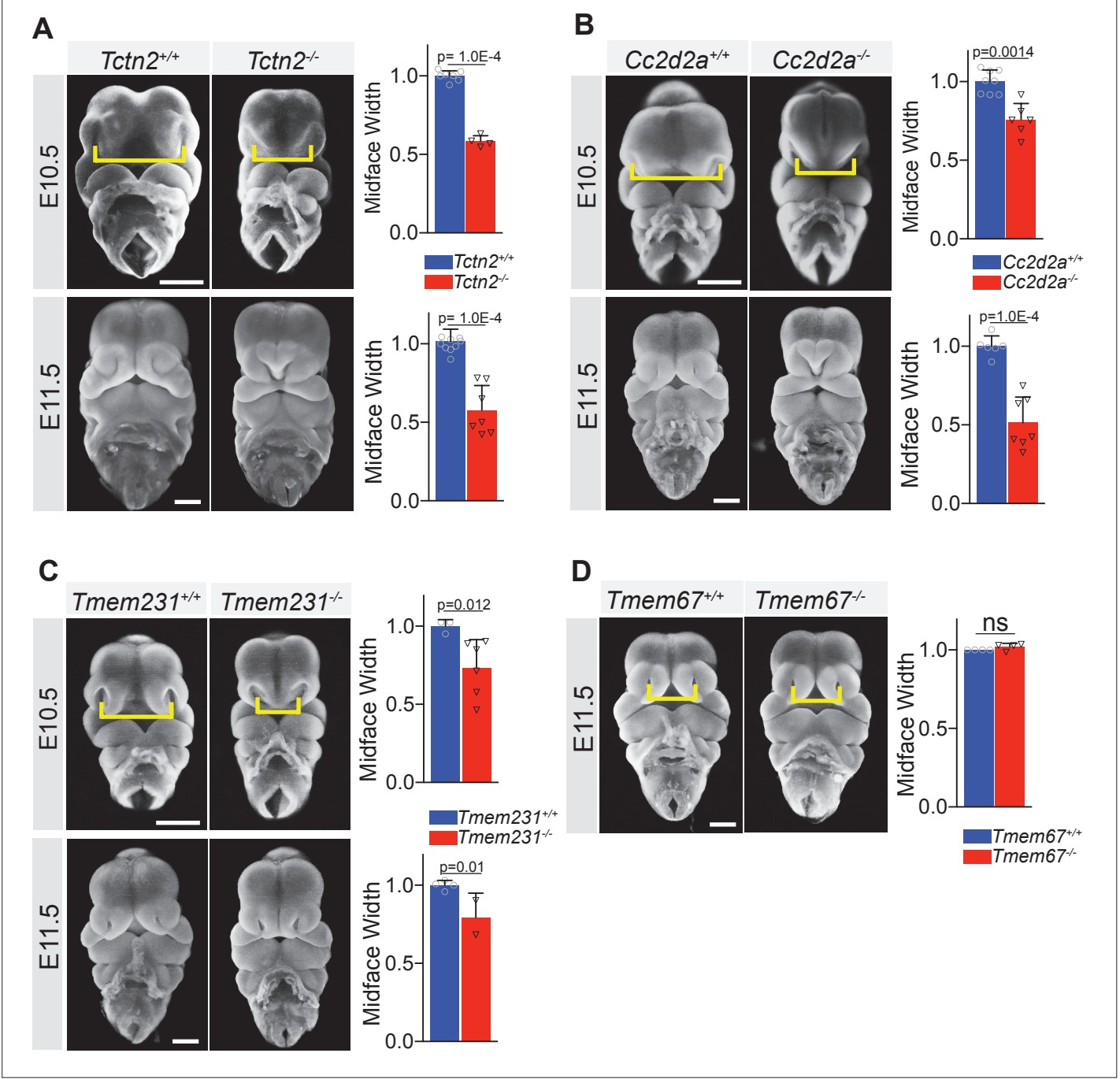

**Figure 1.** The ciliary Meckel syndrome (MKS) transition zone complex is essential for midline facial development. Frontal view images of *Tctn2* (**A**), *Cc2d2a* (**B**), *Tmem231* (**C**) wild-type and null embryos at embryonic day (E)10.5 and E11.5. *Tmem67* null embryos (**D**) display normal midface width at E11.5. Quantification of midface width (denoted by yellow brackets) at respective timepoints was measured via one-way ANOVA followed by Tukey's multiple comparisons test. Data are expressed as mean, and error bars represent the standard deviation (SD) with individual data points (N) representing biological replicates (biologically distinct samples). Scale bar indicates 500 µm. ns = not significant.

The online version of this article includes the following figure supplement(s) for figure 1:

**Figure supplement 1.** The ciliary Meckel syndrome (MKS) transition zone complex is essential for midline facial development.

**Figure supplement 2.** Removing TCTN2 in the neural crest does not result in hypotelorism.

crest can account for the midline hypoplasia observed in *Tctn2* mutants. More specifically, we generated E11.5 *Wnt1*cre;*Tctn2*fl/- embryos and quantified midface width (*Figure 1—figure supplement 2A*). *Wnt1*cre induces recombination throughout the neural crest, and conditional deletion of *Tctn2* in the neural crest abrogated ARL13B localization to cilia (*Figure 1—figure supplement 2B*,C). Interestingly, removing TCTN2 from the neural crest did not cause hypotelorism. These results indicate that altered transition zone function in the neural crest is not the etiology of midline hypoplasia. Therefore, we investigated functions of TCTN2 in the prechordal plate, an early organizing center also critical for the development of anterior head structures.

## *Tctn2* mutants exhibit defects in prechordal plate differentiation soon after gastrulation

As *Tctn2*, *Cc2d2a*, and *Tmem231* mutants all displayed facial midline defects at midgestation, we hypothesized that they shared a role in a patterning event early in craniofacial development. One organizing center critical for early forebrain and craniofacial development is the prechordal plate (*Camus et al., 2000*; *Kiecker and Niehrs, 2001*; *Muenke and Beachy, 2000*; *Rubenstein and Beachy, 1998*; *Som et al., 2014*). The prechordal plate is the anterior-most midline mesendoderm, immediately anterior to the notochord and in contact with the overlying ectoderm. The homeobox gene *Goosecoid* (*Gsc*) is specifically expressed in the prechordal plate at E8.0 and is a marker of differentiation in this organizing center (*Belo et al., 1998*; *Izpisúa-Belmonte et al., 1993*). In contrast, *Shh* and *Brachyury* (*T*) are expressed at E8.0 in both the prechordal plate and notochord and are critical for prechordal plate induction (*Aoto et al., 2009*). Previous work demonstrated that surgical removal of the rat prechordal plate results in midface defects (*Aoto et al., 2009*) that seemed similar to those of the mouse *Tctn2*, *Cc2d2a*, and *Tmem231* mouse mutants.

Therefore, we analyzed the prechordal plate of *Tctn2* mutants by examining the expression of prechordal plate differentiation marker *Gsc* and induction markers *Shh* and *T*. *Gsc* is expressed specifically in the prechordal mesoderm, while *Shh* and *T* are expressed in both the prechordal mesoderm and notochord (*Dale et al., 1997*; *Herrmann, 1991*; *Schulte-Merker et al., 1994*). In situ hybridization analysis revealed that in *Tctn2* mutants, *Shh* and *T* expression in the prechordal plate and notochord were unaffected (*Figure 2A*). Therefore, TCTN2 is not essential for prechordal plate specification. In contrast, *Tctn2* mutants exhibited abrogated *Gsc* expression in the prechordal plate (*Figure 2B*), indicating that TCTN2 is critical for prechordal plate differentiation.

The transition zone is critical for HH signaling, and one HH protein, SHH, is essential for *Gsc* expression in the prechordal plate (*Aoto et al., 2009*; *Chih et al., 2011*; *Garcia-Gonzalo et al., 2011*). Therefore, we investigated whether TCTN2 is required for HH signaling in the prechordal plate by examining the expression of the transcriptional target *Gli1*. *Tctn2* mutants exhibited reduced *Gli1* expression throughout the axial mesendoderm, including the prechordal plate (*Figure 2B*). These results indicate that TCTN2 is dispensable for the formation of the prechordal plate, but is required for midline signaling by SHH to induce *Gsc* expression in this organizing center.

TCTN2 and other members of the MKS complex are required for proper cilia formation in some tissues but not in others (*Garcia-Gonzalo et al., 2011*). Therefore, we examined whether TCTN2 is required for ciliogenesis in the prechordal plate. The prechordal plate expresses FOXA2 (*Jin et al., 2001*; *Figure 2C*). Co-immunostaining embryos for FOXA2 and acetylated tubulin (TUB^Ac), a marker of cilia, revealed that *Tctn2* mutants did not display decreased ciliogenesis in the E8.5 prechordal plate (*Figure 2C*, middle panel). Previous studies have shown that *Tctn1* is expressed in the ventral epithelium of the node of a six-somite stage embryo and in the neural tube, notochord, gut epithelium, and somites at E9.5 (*Reiter, 2006*). *Tctn2* is similarly broadly expressed during mouse development (*Diez-Roux et al., 2011*; *Magdaleno et al., 2006*), and *CC2D2A* is broadly expressed during human development (*Mougou-Zerelli et al., 2009*).

In cell types in which the MKS complex is dispensable for ciliogenesis, like neural progenitors, it is required for localization of ARL13B to cilia (*Garcia-Gonzalo et al., 2011*). Therefore, we examined ARL13B localization in E8.0 control embryos and *Tctn2* mutants and discovered that ARL13B localization to prechordal plate cilia was attenuated without TCTN2 (*Figure 2C*, bottom panel). Thus, TCTN2 is not required for ciliogenesis in the prechordal plate, but does control ciliary composition.

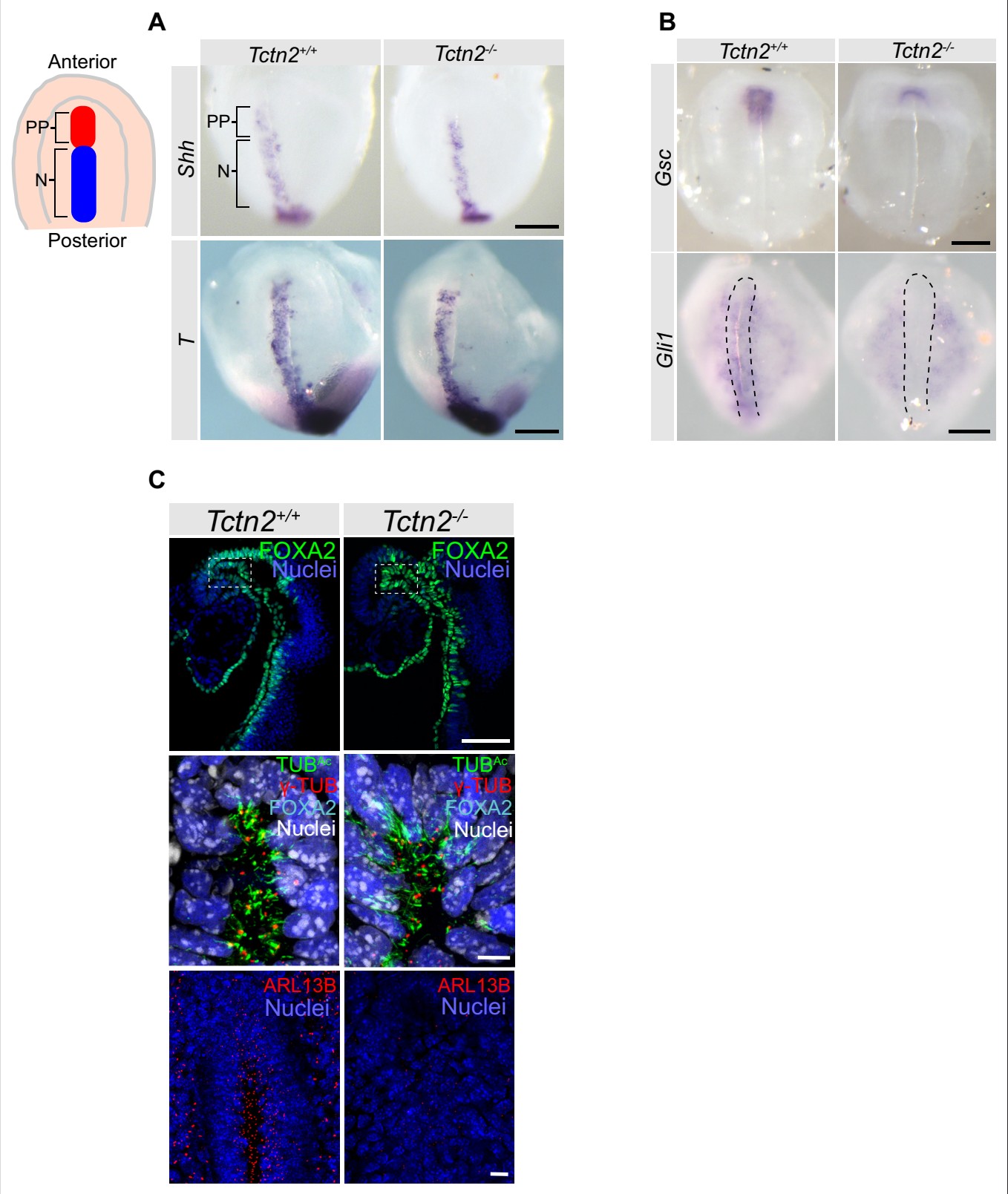

**Figure 2.** *Tctn2* mutants exhibit defects in prechordal plate differentiation soon after gastrulation. (**A**) Whole mount in situ hybridization (WM-ISH) of embryonic day (E)8.0 embryos for axial mesendoderm markers *Shh* and *Brachyury* (*T*). (**B**) WM-ISH of E8.0 embryos for prechordal plate marker *Goosecoid* (*Gsc*) and Hedgehog (HH) pathway target *Gli1*. (**C**) Whole mount immunofluorecence staining for the prechordal plate transcription factor FOXA2, cilia marker acetylated tubulin (TUB^Ac), basal body marker gamma tubulin ($\gamma$-TUB), and ciliary membrane protein ARL13B in E8.0 embryos.

*Figure 2 continued on next page*

*Figure 2 continued*

Middle panel in C is magnified region in top panel highlighted by dotted rectangle and rotated 90 degrees. Scale bar in A–B indicates 0.2 mm, C (top panel) is 100 µm, C (middle and bottom panels) is 10 µm.

## *Tctn2* mutants display decreased HH signaling in the ventral telencephalon

The axial mesendoderm helps pattern the overlying neurectoderm (*Anderson and Stern, 2016*; *Rubenstein and Beachy, 1998*). In the rostral embryo, the prechordal plate patterns the overlying ventral telencephalon via SHH (*Chiang et al., 1996*; *Xavier et al., 2016*). As extirpation of the prechordal plate results in decreased SHH activity in the basal telencephalon (*Aoto et al., 2009*; *Aoto and Trainor, 2015*), we investigated whether the prechordal plate defects observed in *Tctn2* mutants result in mispatterning of the ventral telencephalon. Although *Shh* expression in the notochord was unaffected in *Tctn2* mutants at E8.75, it was severely reduced in the ventral telencephalon (*Figure 3A*). This reduced expression of *Shh* in the ventral telencephalon persisted at E9.5 (*Figure 3A*). These results are concordant with previous findings that mutations in genes encoding other transition zone

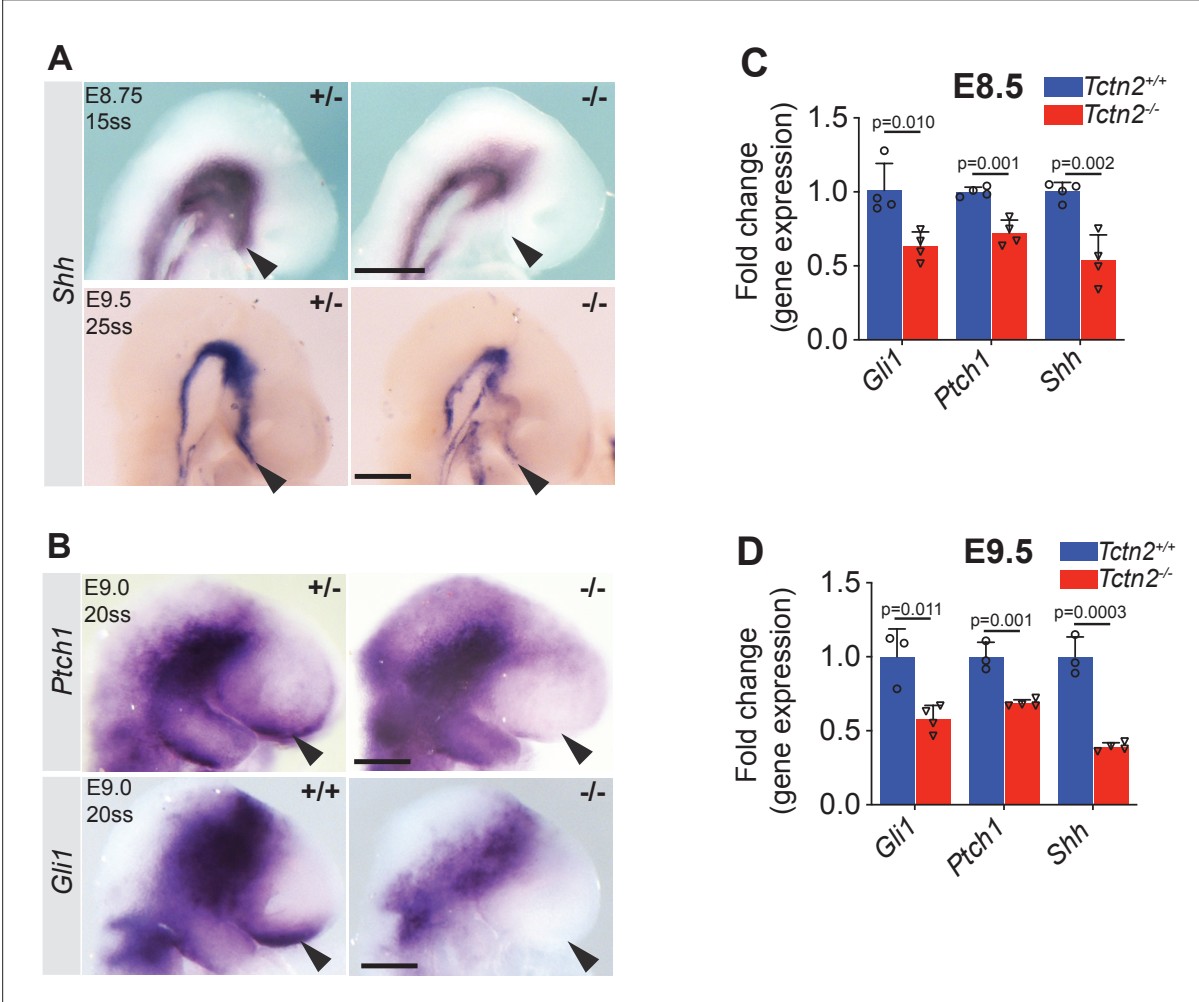

**Figure 3.** *Tctn2* mutants display decreased Hedgehog (HH) signaling in the ventral telencephalon. Whole mount in situ hybridization (WM-ISH) for *Shh* in *Tctn2* control and mutant embryos at embryonic day (E)8.75 and E9.5 (**A**) show reduced expression in the ventral telencephalons (arrowheads) of mutant embryos . WM-ISH for HH pathway targets *Ptch1* and *Gli1* also show reduced expression in the ventral telencephalon in *Tctn2* mutants (**B**, arrowheads). Quantitative real-time polymerase chain reaction (RT-qPCR) analysis of RNA transcripts isolated from E8.5 and E9.5 *Tctn2* control and mutant heads (**C** and **D**, respectively) show reduced levels of *Gli1*, *Ptch1*, and *Shh* transcripts in the absence of TCTN2, consistent with WM-ISH results. Data in C, D represent mean, and error bar represents the standard deviation (SD). Sample size in C and D indicated with 3-4 biological replicates per assay. Student's *t* test performed for statistical analysis. Scale bar indicates 0.5 mm.

components disrupt brain development, resulting in holoprosencephaly, reduced telencephalon size, or exencephaly (*Dowdle et al., 2011*; *Garcia-Gonzalo et al., 2011*; *Reiter, 2006*).

To assess whether HH pathway activity was compromised by the absence of TCTN2, we assessed the expression of the HH transcriptional targets *Gli1* and *Ptch1*. Whole mount in situ hybridization (WM-ISH) of *Tctn2* mutants revealed dramatically reduced or absent expression of both *Gli1* and *Ptch1*, especially in the basal forebrain (*Figure 3B*). Consistent with the WM-ISH data, quantitative real-time polymerase chain reaction (RT-qPCR) analysis of E8.5 (*Figure 3C*) and E9.5 (*Figure 3D*) *Tctn2* mutant heads also revealed decreased expression of *Shh*, *Ptch1*, and *Gli1*, revealing that *Tctn2* mutants exhibit both an early defect in prechordal plate differentiation and a defect in HH signaling in the adjacent neurectoderm.

## TCTN2 protects neurectoderm and facial ectoderm from apoptosis

In the developing craniofacial complex, SHH induces cell proliferation (*Hu and Helms, 1999*; *Hu et al., 2015*). Therefore, we assessed if the reduction in facial midline width in *Tctn2* mutants was due to decreased cell proliferation. More specifically, we measured cell proliferation by quantitating phospho-histone H3 (pHH3) in the components of the craniofacial complex – the forebrain, hindbrain, facial ectoderm, and mesenchyme (*Figure 4A and B*). *Tctn2* mutants showed no differences in amount or spatial distribution of cell proliferation.

In other developmental contexts, HH signaling promotes cell survival (*Ahlgren and Bronner-Fraser, 1999*; *Aoto et al., 2009*; *Aoto and Trainor, 2015*; *Litingtung and Chiang, 2000*). Therefore, we assessed apoptosis in the craniofacial complex of *Tctn2* mutants. Quantification of TUNEL staining revealed that apoptosis was increased in the neurectoderm, facial ectoderm, and mesenchyme of *Tctn2* mutants compared to controls, and most dramatically in the ventral telencephalon (*Figure 4C and D*). To further test whether apoptosis is increased in the absence of TCTN2, we stained for activators of the intrinsic apoptotic pathway, cleaved-caspase-3 and caspase-9 (activated CASP3 and CASP9). Both activated CASP3 and CASP9 were increased in the ventral telencephalon, facial ectoderm, and mesenchyme of *Tctn2* mutants at E9.5 (*Figure 4E*). These data indicate that TCTN2 is required to protect against cell death, but does not affect proliferation, in the neurectoderm, non-neural ectoderm and neural crest mesenchyme. As SHH also protects neurectoderm from apoptosis (*Thibert et al., 2003*), we propose that TCTN2 mediates cell survival by promoting HH signaling and that the increase in cell death in the neurectoderm, mesenchyme and facial ectoderm underlies the midline hypoplasia in transition zone mutants.

Our finding that selective disruption of transition zone function in the neural crest did not contribute to midline growth (*Figure 1—figure supplement 2*) coupled with increased apoptosis in the *Tctn2* mutant, neurectoderm and facial ectoderm (*Figure 4C–D*) led us to investigate where TCTN2 is required to coordinate midline facial development. *Isl1*<sup>Cre</sup> induced robust reporter recombination in the prechordal plate at E8.5 (*Figure 4—figure supplement 1A*) and *Sox1*<sup>Cre</sup> induced recombination in the neuroectoderm by E9.5 (*Figure 4—figure supplement 1B*). Using *Isl1*<sup>Cre</sup> and *Sox1*<sup>Cre</sup>, we removed *Tctn2* from the prechordal plate or neurectoderm, respectively. Both *Isl1*<sup>Cre</sup>;*Tctn2*<sup>fl/-</sup> and *Sox1*<sup>Cre</sup>;*Tctn2*<sup>fl/-</sup> embryos displayed no decrease in midline width at E11.5 (*Figure 4—figure supplement 1C and D*).

Similarly, we investigated the function of TCTN2 in the facial ectoderm and telencephalon. *Tcfap2a*<sup>Cre</sup> and *Foxg1*<sup>Cre</sup> activated recombination in the facial ectoderm or telencephalon and ectoderm, respectively (*Figure 4—figure supplement 1E*, F). Both *Tcfap2a*<sup>Cre</sup>;*Tctn2*<sup>fl/-</sup> and *Foxg1*<sup>Cre</sup>;*Tctn2*<sup>fl/-</sup> embryos displayed no decrease in midline width at E11.5 (*Figure 4—figure supplement 1G*, H).

Analysis of ARL13B localization to cilia at E8.5 and E9.5 in the prechordal plate and neurectoderm in *Isl1*<sup>Cre</sup>;*Tctn2*<sup>fl/-</sup> and *Sox1*<sup>Cre</sup>;*Tctn2*<sup>fl/-</sup> embryos, respectively, revealed that there was no alteration in ciliogenesis or ciliary localization of ARL13B (*Figure 4—figure supplement 2A and B*). Analysis of ciliary ARL13B localization in the ectoderm of E11.5 *Tcfap2a*<sup>Cre</sup>;*Tctn2*<sup>fl/-</sup> mutants similarly revealed persistent ARL13B localization to cilia (*Figure 4—figure supplement 2C*). In contrast, *Foxg1*<sup>Cre</sup>;*Tctn2*<sup>fl/-</sup> mutants exhibited loss of ARL13B ciliary localization in the basal telencephalon at E11.5 (*Figure 4—figure supplement 2D*). These results indicate that *Tctn2* conditional deletion in either the prechordal plate, neurectoderm, facial ectoderm, or forebrain individually fails to recapitulate the midline narrowing observed in *Tctn2*<sup>-/-</sup> embryos. However, the persistence of ARL13B at cilia suggests that TCTN2 function persists for some time after recombination or TCTN2 functions in the prechordal plate, neurectoderm and facial ectoderm may be important for facial development.

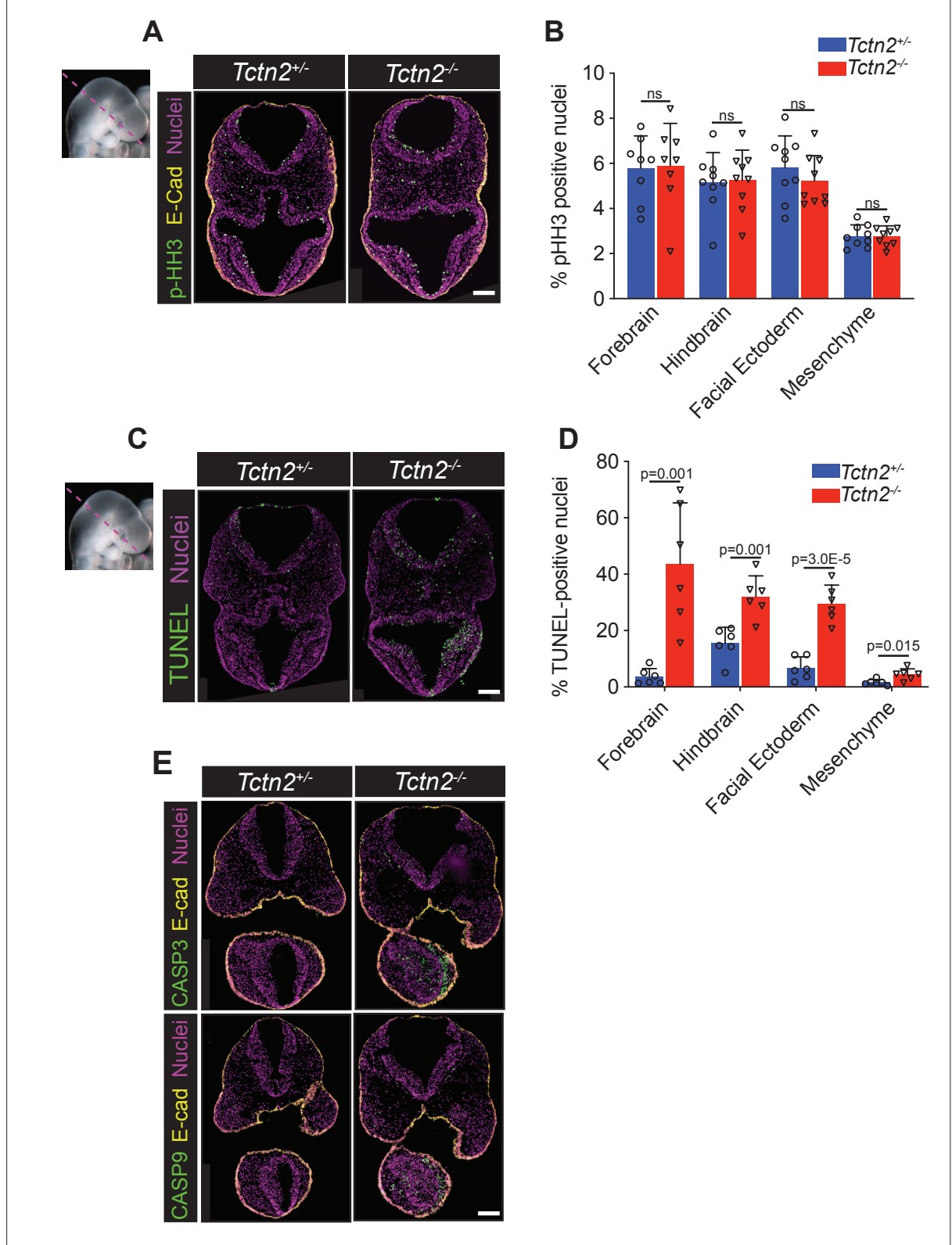

**Figure 4.** TCTN2 protects the neurectoderm and facial ectoderm from apoptosis. Immunostaining for proliferation marker phospho-histone H3 (pHH3) in transverse sections of *Tctn2* control and mutant embryonic day (E9.5) embryos (**A**) with corresponding quantification (**B**). (**C**) TUNEL staining for analysis of apoptosis in *Tctn2* E9.5 embryos with corresponding quantification (**D**). (**E**) Immunostaining for intrinsic apoptotic pathway components cleaved-caspase-3 and cleaved-caspase-9 in *Tctn2* control and mutant E9.5 embryos. Quantified data represent N = 3 biological replicates (biologically

*Figure 4 continued on next page*

*Figure 4 continued*

distinct samples) with a minimum of two sections analyzed per sample. Student's *t* test performed for statistical analysis. Data in B, D represent the mean, and error bars represent the standard deviation (SD). Scale bar indicates 100 μm. ns = not significant.

The online version of this article includes the following figure supplement(s) for figure 4:

**Figure supplement 1.** Deletion of *Tctn2* in the prechordal plate by *Isl1*[Cre], neurectoderm by *Sox1*[Cre], facial ectoderm by *Tcfap2a*[Cre], or forebrain and facial ectoderm by *Foxg1*[Cre] does not result in hypotelorism.

**Figure supplement 2.** Residual TCTN2 function through persistent ARL13B ciliary localization in TCTN2 conditional mutants.

## Reducing *Ptch1* gene dosage rescues the facial midline defect in transition zone mutants

To assess whether decreased HH signaling is not just correlated with midfacial hypoplasia in transition zone mutants, but is causative, we investigated whether modulating the HH pathway could rescue the midface defects. We employed a strategy targeting *Ptch1*, a negative regulator of the HH pathway, by generating *Tctn2*$^{-/-}$;*Ptch1*$^{+/-}$ embryos and comparing them to *Tctn2*$^{-/-}$;*Ptch1*$^{+/+}$ embryos. Surprisingly, removing a single allele of *Ptch1* in *Tctn2* mutants restored midface width at E11.5 (*Figure 5A and B*).

To assess whether removing a single allele of *Ptch1* restores midface width in other transition zone mutants, we generated *Cc2d2a*$^{-/-}$;*Ptch1*$^{+/-}$ and *Cc2d2a*$^{-/-}$;*Ptch1*$^{+/+}$ embryos. As with *Tctn2*, removing a single allele of *Ptch1* restored midface width in *Cc2d2a* mutants (*Figure 5C and D*). Thus, reducing *Ptch1* gene dosage rescues midface expansion in both models of ciliopathy-associated hypoplasia.

As we had hypothesized that increased apoptosis underlay the midface hypoplasia of *Tctn2*$^{-/-}$;*Ptch1*$^{+/+}$ embryos, we assessed apoptosis in *Tctn2*$^{-/-}$;*Ptch1*$^{+/-}$ embryos via TUNEL staining. *Tctn2*$^{-/-}$;*Ptch1*$^{+/-}$ exhibited less apoptosis than *Tctn2*$^{-/-}$;*Ptch1*$^{+/+}$ embryos, with a restriction of apoptosis in the ventral telencephalon (*Figure 5E–F*). These results bolster the hypothesis that increased midline apoptosis accounts for the midline hypoplasia of transition zone mutants.

As genes encoding transition zone MKS components are epistatic to *Ptch1* (*Reiter, 2006*), we pondered how reducing *Ptch1* gene dosage restored facial midline development to transition zone MKS component mutants. The best studied role for PTCH1 is in repression of the HH signal transduction pathway. Therefore, we examined HH signal transduction pathway activity in *Tctn2*$^{-/-}$;*Ptch1*$^{+/+}$ and *Tctn2*$^{-/-}$;*Ptch1*$^{+/-}$ embryos. WM-ISH revealed that expression of neither *Shh* nor *Gli1* was increased in the ventral telencephalons of *Tctn2*$^{-/-}$;*Ptch1*$^{+/-}$ in comparison to *Tctn2*$^{-/-}$;*Ptch1*$^{+/+}$ embryos (*Figure 5G*). Thus, the restoration of midface growth by reduction of *Ptch1* gene dosage is not due to a restoration of HH signal transduction.

In addition to its role in regulating HH signal transduction, PTCH1 exhibits pro-apoptotic activity in vitro (*Thibert et al., 2003*). In the absence of ligand, PTCH1 C-terminal domain is cleaved and binds scaffolding proteins TUCAN1 and DRAL to recruit caspase-9 and activate caspase-3, resulting in apoptosis. The colocalization of another PTCH1-binding protein that regulates apoptosis, X-linked inhibitory apoptosis protein (XIAP), with PTCH1 at cilia (*Aoto and Trainor, 2015*), raises the possibility that PTCH1 cleavage occurs at the primary cilium to induce apoptosis. In addition, PTCH1 may induce apoptosis at the plasma membrane. As reducing *Ptch1* gene dosage reduces apoptosis without increasing HH signal transduction in *Tctn2* mutants, our data supports a model in which the PTCH1-mediated death of the midline neurectoderm, facial ectoderm, and neural crest-derived mesenchyme, and not alterations in HH signal transduction within those cells, is the etiology of midface defects in transition zone mutants. Taken together, these results suggest a working model for how ciliary HH signaling regulates midface development.

In wild-type embryos, HH signaling within the prechordal plate is critical for *Gsc* expression and the induction of *Shh* in the adjacent neurectoderm and inhibition of apoptosis (*Figure 6A*). In transition zone mutants, defects in prechordal plate signaling cause reduced SHH in the neurectoderm, resulting in increased PTCH1-mediated cell death and midline collapse (*Figure 6B*). In transition zone mutants lacking a single allele of *Ptch1*, reduced SHH in the neurectoderm persists, but the attenuated PTCH1 is no longer sufficient to induce extensive cell death, allowing for normal midline facial development (*Figure 6C*).

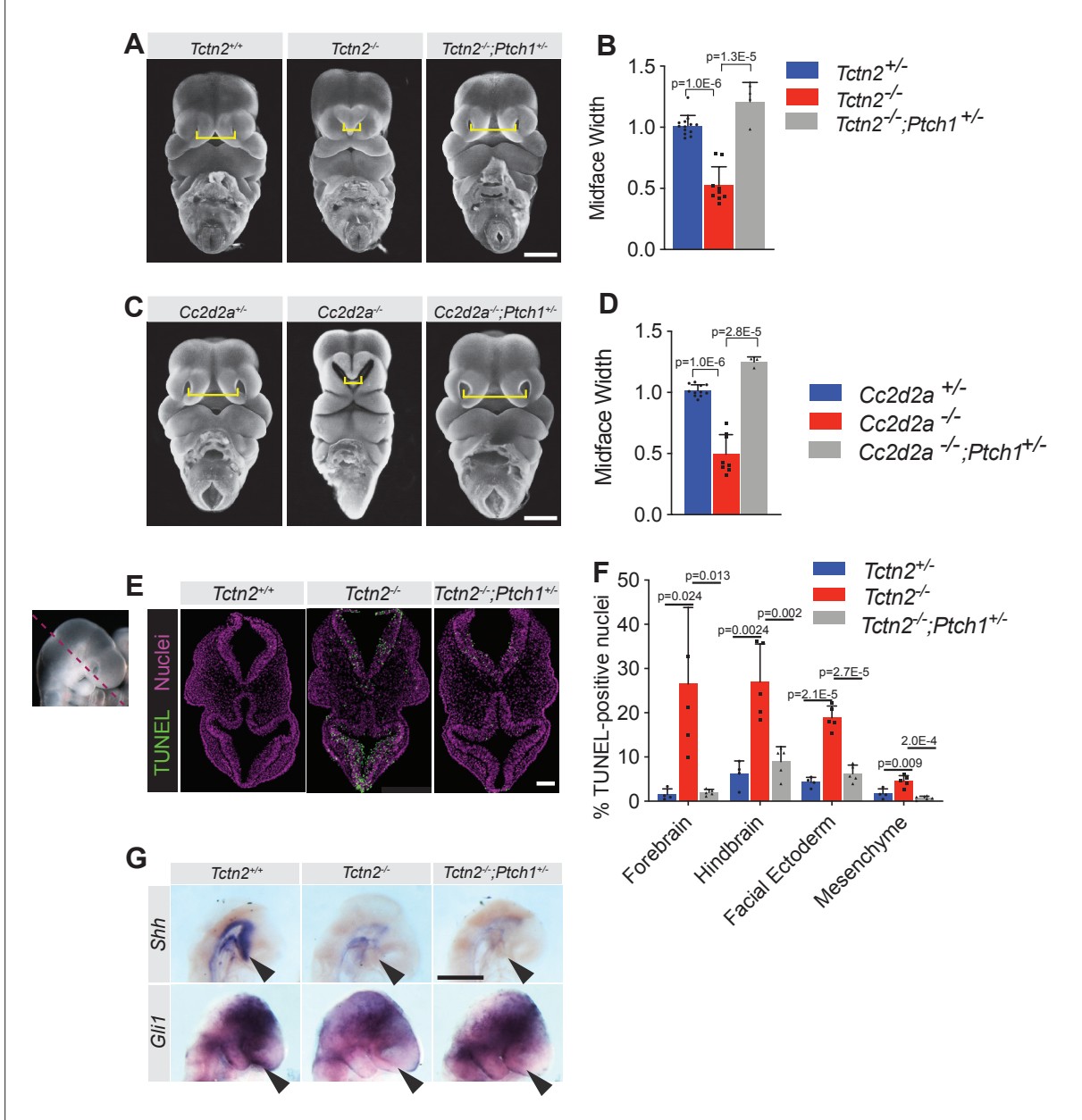

**Figure 5.** Reducing *Ptch1* gene dosage rescues the facial midline defect in transition zone mutants. (**A**) Frontal images of *Tctn2* wild-type, mutant, and *Ptch1⁺/⁻* rescue embryonic day (E)11.5 embryos with corresponding midline width quantification (**B**). (**C**) Frontal images of *Cc2d2a* wild-type, mutant, and *Ptch1⁺/⁻* rescue E11.5 embryos with corresponding midline width quantification (**D**). (**E**) TUNEL assay sections of E9.5 *Tctn2* wild-type, mutant, and *Ptch1⁺/⁻* rescue with corresponding quantification (**F**). (**G**) Whole mount in situ hybridization (WM-ISH) of E9.5 *Tctn2* wild-type, mutant, and *Ptch1⁺/⁻* rescue embryos for *Shh* and *Gli1*. Quantified data represent N = 3 biological replicates (biologically distinct samples) with a minimum of two sections analyzed per sample. Data in B, D, F represent the mean, and error bars represent the standard deviation (SD). For statistical analysis, one-way ANOVA was performed with Tukey's multiple comparison test. Scale bars indicate 500 µm (**A, C, G**) and 100 µm (**E**).

## Discussion

Using a combination of genetic, developmental, and biochemical techniques, we have identified a mechanism by which disruption of MKS transition zone proteins (TCTN2, CC2D2A, and TMEM231) results in midline hypoplasia and hypotelorism. We traced the origin of the molecular defect contributing to the midline phenotype to the prechordal plate, defects in which resulted in reduced HH pathway activation and cell survival in the adjacent neurectoderm and facial midline collapse. We uncovered *Ptch1* gene dosage as a key mediator of cell survival in the facial midline of transition zone

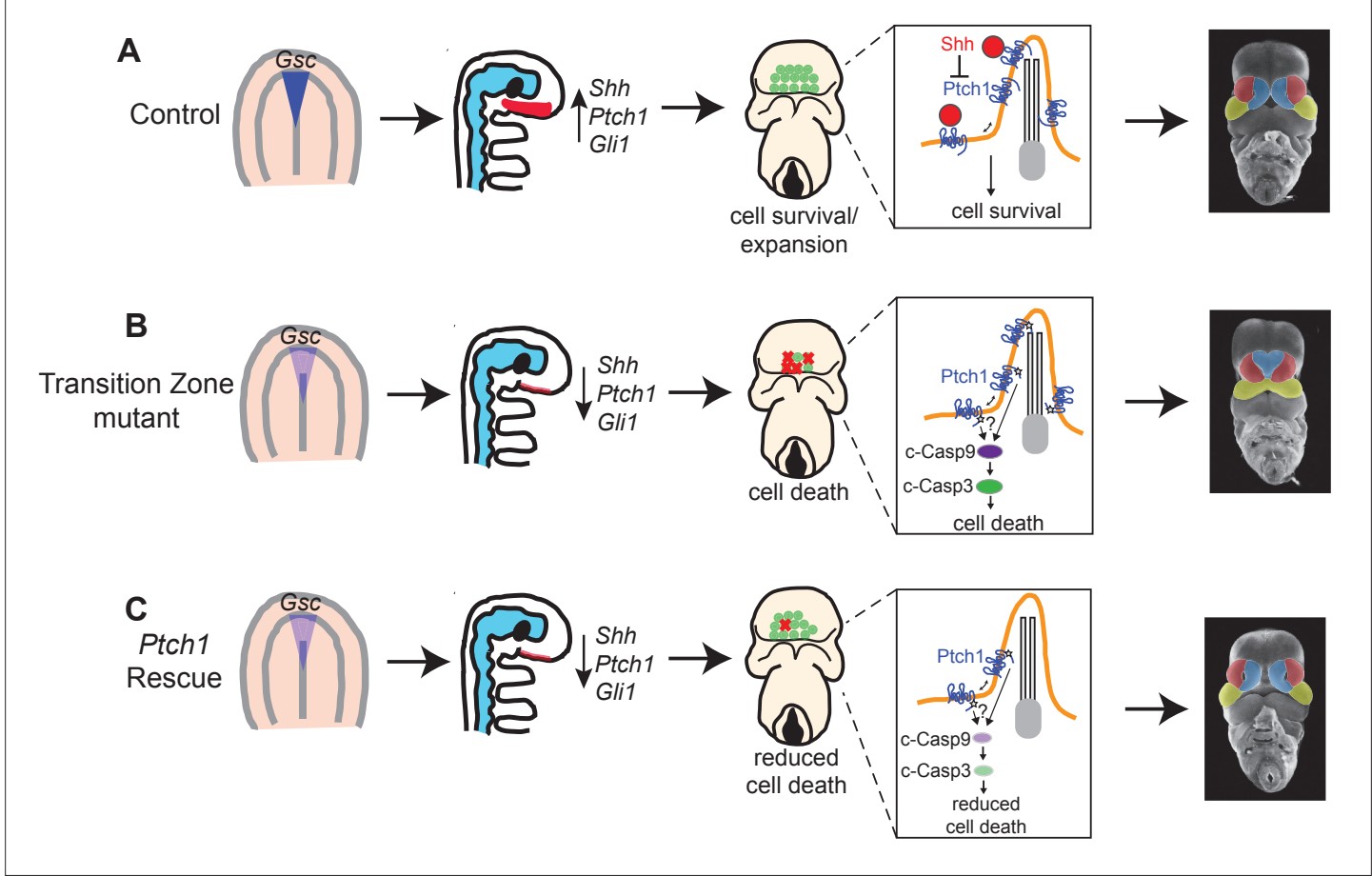

**Figure 6.** Model for transition zone coordination of facial midline development. (**A**) In wild-type embryos, the transition zone complex mediates signaling in the prechordal plate and Hedgehog (HH) pathway activation in the adjacent neurectoderm to at allow for cell survival and normal midline development. (**B**) In transition zone mutants, disrupted signaling in the prechordal plate results in reduced *Shh* and HH pathway activation in the neurectoderm resulting in increased cell death and corresponding collapse of the facial midline. (**C**) In transition zone mutants with *Ptch1* haploinsufficiency (*Ptch1* rescue), reduced cell death allows for normal midface development despite persistent reduction of *Shh* and HH pathway activation in the neurectoderm.

mutants, as loss of a single allele of *Ptch1* rescued cell survival and midline development in *Tctn2* and *Cc2d2a* mutants. Together, these results reveal a new paradigm whereby primary cilia mediate signal crosstalk from the prechordal plate to the adjacent neurectoderm to promote cell survival, without which the facial midline collapses and hypotelorism results.

Different ciliopathies are associated with either narrowing or expansion of the facial midline (hypotelorism or hypertelorism) (*Schock et al., 2015*; *Zaghloul and Brugmann, 2011*). Severe ciliopathies associated with perinatal lethality, such as MKS, can present with hypotelorism or hypertelorism while other ciliopathies such as Joubert syndrome typically present with hypertelorism (*Brugmann et al., 2010b*; *MacRae et al., 1972*; *Schock and Brugmann, 2017*). How disruption of ciliary functions can give rise to these opposing phenotypes has been an active area of interest. Hypertelorism is attributable to roles for cilia in promoting GLI3 repressor formation in neural crest cells (*Brugmann et al., 2010a*; *Chang et al., 2014*; *Chang et al., 2016*; *Liu et al., 2014*). Our work implicates a distinct etiology of hypotelorism: rather than involving neural crest, midline hypoplasia can be caused by defects in the ciliary transition zone in the prechordal plate.

The earliest alteration we detected in *Tctn2*[-/-] signaling centers that regulate craniofacial development was in the prechordal plate at the end of gastrulation. TCTN2 was dispensable for the expression of *Shh* and *T* in the prechordal plate, indicating that induction and specification of the prechordal plate were not dependent on transition zone function. In contrast, TCTN2 was essential for prechordal plate expression of *Gli1* and *Gsc*. In tissues such as the limb bud, TCTN2 is dispensable for ciliogenesis

but critical for ciliary HH signaling and the induction of HH target genes such as *Gli1* (*Chih et al.,* *2011*; *Dowdle et al., 2011*; *Garcia-Gonzalo et al., 2011*). We found that, similarly in the prechordal plate, TCTN2 is dispensable for ciliogenesis but critical for induction of *Gli1*.

In many developmental events, such as limb patterning, SHH signals to neighboring cells to induce a pattern (*Panman and Zeller, 2003*; *Zhulyn et al., 2014*). In other developmental events, such as notochord to neural tube signaling, SHH signals produced by the notochord induce the expression of *Shh* in the overlaying neural tube (*Fuccillo et al., 2006*). SHH produced by the prechordal plate may fall into the latter category, as the absence of *Shh* expression in *Tctn2-/-* embryos presages reduced *Shh* expression and reduced expression of HH pathway transcriptional targets *Gli1* and *Ptch1* in the region of the basal forebrain sometimes referred to as the rostral diencephalon ventral midline. Thus, in the posterior midline, the notochord induces *Shh* expression in overlying neuroectoderm, and in the anterior midline, the prechordal plate induces *Shh* expression in the overlying neuroectoderm. In the failure of the prechordal plate to induce Shh expression in the overlying neurectoderm, *Tctn2* mutants recapitulate previous observations of *Lrp2* mutants which display attenuated responses to SHH (*Christ et al., 2012*). One possible mechanism by which SHH may activate expression of *Shh* in the basal forebrain is via the induction of the transcription factor SIX3. SIX3 is regulated by HH signaling and required for the induction of *Shh* in the developing forebrain (*Geng et al., 2008*; *Jeong et al., 2008*). However, our observation that *Six3* expression is unaltered in the forebrains of *Tctn2* mutants diminishes support for this hypothesis.

In caudal neural tube and limb patterning, HH signals induce patterning. In hair follicles and cerebellar granule cells, HH signaling promotes proliferation. In addition to roles in patterning and proliferation, HH signals can bind to PTCH1 to inhibit apoptosis (*Aoto and Trainor, 2015*; *Borycki et al.,* *1999*; *Thibert et al., 2003*). Our data are consistent with a role of PTCH1 in promoting apoptosis in the neuroectoderm and facial ectoderm which is inhibited by prechordal plate-produced SHH. In the absence of TCTN2, the ventral telencephalon does not produce SHH, releasing PTCH1 to promote apoptosis in the midline and resulting in midface hypoplasia. This model is consistent with previous data demonstrating that surgical ablation of the prechordal plate reduces the forebrain (*Aoto et al.,* *2009*). Increased cleaved-caspase-3 and caspase-9 staining in the basal forebrain and facial ectoderm of E9.5 *Tctn2* mutants provides further support for apoptosis contributing to the associated midline hypoplasia.

We speculate that in tissues, such as the developing spinal cord where cilia are required for SHH expression in the floor plate, ciliary dysfunction will cause increased apoptosis. In other tissues, such as the limb bud where cilia are not required for SHH expression in the zone of proliferating activity, we predict that ciliary dysfunction will not cause increased apoptosis. Thus, ciliary dependence of SHH expression may etermine which tissues, like the craniofacial midline, increase apoptosis upon ciliary dysfunction.

To try to narrow down the tissues in which transition zone function is critical for midline facial development, we used conditional mouse genetics to delete *Tctn2* in different tissues that comprise the craniofacial complex. Deletion of *Tctn2* in the prechordal plate (via *Isl1*[Cre]) or the neurectoderm (via *Sox1*[Cre]) did not recapitulate the midline hypoplasia observed in *Tctn2-/-* embryos (*Figure 4—figure* *supplement 1*). Similarly, deletion of *Tctn2* in the facial ectoderm (via *Tcfap2a*[Cre]) or in the forebrain and facial ectoderm (via *Foxg1*[Cre]) also failed to result in midline hypoplasia (*Figure 4—figure supplement 1*). These results could reflect the dispensibility of TCTN2 in these tissues for facial development, or could reflect residual TCTN2 function after *Tctn2* recombination. This latter possibility is supported by persistent ARL13B localization at the cilium in several tissues after conditional deletion of *Tctn2*. As the half-life of TCTN2 and the TCTN2 level required to sustain transition zone activity are unclear, residual TCTN2 activity may support normal ciliary signaling even after conditional gene ablation.

Surprisingly, removing one allele of *Ptch1* fully rescues the midface defect in both *Tctn2* and *Cc2d2a* transition zone cilia mutants. Even more surprisingly, this phenotypic rescue is not associated with restoration of either *Shh* expression or HH pathway activation in the basal forebrain. We propose that reducing *Ptch1* levels attenuates the PTCH1-mediated pro-apoptotic program normally attenuated by SHH in the basal forebrain. However, it remains possible that reducing PTCH1 levels activates GLI effectors to induce an anti-apoptotic program that does not include general HH target genes or *Shh*.

In summary, we have identified the primary cilia transition zone as a critical regulator of facial midline development. The transition zone component TCTN2 was critical for SHH signaling in the prechordal

plate and uncovered a signaling paradigm whereby the transition zone promotes cell survival by mediating crosstalk between the prechordal plate and neurectoderm to promote HH pathway activation. These results provide insights into how primary cilia mediate cell survival to promote facial development.

# Materials and methods

## Key resources table

| Reagent type (species) or resource | Designation | Source or reference | Identifiers | Additional information |
|---|---|---|---|---|
| Antibody | Anti-Arl13b (rabbit polyclonal) | Proteintech | 17711–1-AP | (1:1000) |
| Antibody | Anti-cleaved-caspase-3 (Asp175) (rabbit polyclonal) | Cell Signaling | #9664 | (1:400) |
| Antibody | Anti-cleaved-caspase-9 (Asp353) | Cell Signaling | #9509 | (1:100) |
| Antibody | Anti-phospho-histone H3 (Ser28) (rabbit polyclonal) | Cell Signaling | #9713 | (1:400) |
| Antibody | Anti-acetylated tubulin (mouse monoclonal) | Sigma-Aldrich | T6793 | (1:1000) |
| Antibody | Anti-FoxA2 (rabbit monoclonal) | abcam | Ab108422 | (1:400) |
| Antibody | Anti-Gamma tubulin (goat polyclonal) | Santa Cruz Biotechnology | Sc-7396 | (1:200) |
| Antibody | Anti-GFP (chicken polyclonal) | Aves Labs | GFP-1020 | (1:1000) |
| Antibody | Anti-E-cadherin (rat monoclonal) | Invitrogen | 13–1900 | (1:1000) |
| Commercial assay or kit | RNAeasy Micro Kit | QIAGEN | 74004 | |
| Commercial assay or kit | In Situ Cell Death Detection Kit | Roche | 11684795910 | |
| Commercial assay or kit | iScript cDNA synthesis kit | BioRad | 1708891 | |
| Genetic reagent (*Mus musculus*) | *Tctn2$^{tm1.1Reit}$* | PMID:21565611 | MGI:5292130; RRID:MGI:5292219 | |
| Genetic reagent (*Mus musculus*) | *Cc2d2a$^{Gt(AA0274)Wtsi}$* | PMID:21725307 | MGI:4344514; RRID:MGI:5292228 | |
| Genetic reagent (*Mus musculus*) | *Tmem231$^{Gt(OST335874)Lex}$* | PMID:22179047 | MGI:4284576; RRID:MGI:5301844 | |
| Genetic reagent (*Mus musculus*) | *Tmem67$^{tm1Dgen}$* | PMID:21725307 | MGI:5292220; RRID:MGI:5292226 | |
| Genetic reagent (*Mus musculus*) | *Wnt1$^{Cre}$: H2az2Tg$^{(Wnt1-cre)11Rth}$* | PMID:9843687 | MGI:2386570; RRIDSupplemental:IMSR_JAX:003829 | Gift from Brian Black |
| Genetic reagent (*Mus musculus*) | *Isl1$^{tm1(cre)Sev}$* | PMID:16556916 | MGI:3623159; RRID:IMSR_HAR:3351 | Gift from Brian Black |
| Genetic reagent (*Mus musculus*) | *Sox1$^{tm1(cre)Take}$* | PMID:17604725 | MGI:3807952; RRID:IMSR_RBRC05065 | Gift from Jeff Bush |
| Genetic reagent (*Mus musculus*) | *Foxg1$^{tm1(cre)Skm}$* | PMID:10837119 | MGI:1932522; RRID:IMSR_JAX:004337 | Gift from Stavros Lomvardas |
| Genetic reagent (*Mus musculus*) | *Tg(Tcfap2a-cre)$^{1Will}$* | PMID:21087601 | MGI:4887352; RRID:MGI:4887452 | Gift from Trevor Williams |
| Sequence-based reagent | *Shh* in situ probe | PMID:7916661 | | Gift from Andrew McMahon |
| Sequence-based reagent | *Gsc* in situ probe | PMID:1352187 | | Gift from Edward De Robertis |

*Continued on next page*

Continued

| Reagent type (species) or resource | Designation | Source or reference | Identifiers | Additional information |
|---|---|---|---|---|
| Sequence-based reagent | *Ptch1* in situ probe | PMID:10395791 | | Gift from Lisa Goodrich |
| Sequence-based reagent | RT-qPCR primers | This paper | | *See **Supplementary file 1** |
| Sequence-based reagent | Genotyping primers | This paper | | *See **Supplementary file 1** |
| Software, algorithm | GraphPad Prism | GraphPad Prism (https://graphpad.com) | RRID:SCR_002798 | |
| Software, algorithm | ImageJ | ImageJ (http://imagej.nih.gov/ij/) | RRID:SCR_003070 | |

## Mouse strains

All mouse protocols were approved by the Institutional Animal Care and Use Committee (IACUC) at the University of California, San Francisco. *Tctn2*[+/-] (*Tctn2*[tm1.1Reit]), *Cc2d2a*[+/-] (*Cc2d2a*[Gt(AA0274)Wtsi]), *Tmem231*[+/-] (*Tmem231*[Gt(OST335874)Lex]), and *Tmem67*[+/-] (*Tmem67*[tm1Dgen]) mouse allele references can be found in the Key Resources table (**Garcia-Gonzalo et al., 2011**). *Wnt1*[Cre] (*Tg(Wnt1-cre)*[11Rth]) and *Islet1*[Cre] (*Isl1*[tm1(cre)Sev]) mice were obtained from Brian Black, *Sox1*[Cre] (*Sox1*[tm1(cre)Take]) mice were obtained from Jeff Bush, *Foxg1*[Cre] (*Foxg1*[tm1(cre)Skm]) mice were obtained from Stavros Lomvardas, and *Tcfap2a*[Cre] (*Tg(Tcfap2a-cre)*[1Will]) mice were obtained from Trevor Williams. The *Ptch1*[tm1Mps] allele was used in this study as a null allele. The *Tctn2* conditional allele (*Tctn2*[lox] or *Tctn2*[tm1cReit]) was derived from the *Tctn2*[tm1aReit] allele from which the validated *Tctn2*[tm1.1Reit] null allele was previously derived (**Garcia-Gonzalo et al., 2011**; **Sang et al., 2011**). More specifically, we removed the puromycin resistance cassette of *Tctn2*[tm1aReit] by Cre-mediated recombination, leaving two loxP sites flanking the first three *Tctn2* exons to generate *Tctn2*[fl]. All mice were maintained on a C57BL/6 J background. For timed matings, noon on the day a copulation plug was detected was considered to be 0.5 days postcoitus. Genotyping primers for all mouse strains used in this study can be found in **Supplementary file 1**.

## Immunofluorescence

The antibodies used in this study were rabbit α-ARL13B (1:1000, Proteintech 17711–1-AP), rabbit α-cleaved-caspase-3 (Asp175) (1:400, Cell Signaling #9664), rabbit α-cleaved-caspase-9 (Asp353) (1:100, Cell Signaling #9509), rabbit α-phospho-histone H3 (Ser28) (1:400, Cell Signaling #9713), chicken anti-GFP (1:1000, Aves labs GFP-1020), goat gamma-tubulin (1:200, Santa Cruz sc7396), mouse acetylated-tubulin (1:1000, Sigma T6793), rat E-cadherin (1:1000, Invitrogen 13–1900), and rabbit FoxA2 (1:400 abcam ab108422). The In Situ Cell Death Detection Kit, Fluorescein (Roche) was used for TUNEL cell death assay. For immunofluorescence antibody staining of frozen tissue sections, embryos were fixed overnight in 4 % PFA/PBS, washed in PBS and cryopreserved via overnight incubation in 30 % sucrose/PBS. Embryos were embedded in OCT and frozen at –80 °C. Frozen OCT blocks were cut into 10 µM sections. For immunostaining, frozen sections were washed 3 × 5′ in PBST (0.1%Tween-20/PBS) followed by blocking for 2 hr in blocking solution (5 % donkey serum in PBS + 0.3 % Triton X-100 +0.2 % Na-azide). Slides were incubated overnight in primary antibody diluted in blocking solution at 4 degrees. The following day, slides were washed 3 × 10′ in PBST, stained with appropriate AlexaFluor 488, 568, or 647 conjugated secondary antibodies (Life Technologies) at 1:1000 and Hoecsht or DAPI nuclear stain in blocking buffer for 1 hr, rinsed 3 × 10′ with PBST and mounted using Fluoromount-G (Southern Biotech). All steps performed at room temperature unless otherwise noted. *Note: For gamma-tubulin antibody staining, antigen retrieval by incubating with 1%SDS/PBST for 5 min prior to blocking and primary antibody incubation is required for good staining. Stained samples were imaged on a Leica SP-5 confocal microscope. Images were processed using FIJI (ImageJ).

## In situ hybridization

WM-ISH was performed as previously described (*Harrelson et al., 2012*). DIG-labeled riboprobes were made using plasmids from the following sources: *Shh* (*Echelard et al., 1993*), *Gsc* (*Blum et al., 1992*), *Ptch1* (*Goodrich et al., 1999*), *Foxa2* (*Brennan et al., 2001*), *Gli1* (EST W65013).

## RT-qPCR

For gene expression studies, RNA was extracted from E8.5/E9.5 embryo heads using an RNAeasy Micro Kit (QIAGEN), and cDNA synthesis was performed using the iScript cDNA synthesis kit (BioRad). RT-qPCR was performed using EXPRESS Sybr GreenER 2 × master mix with ROX (Invitrogen) and primers homologous to *Shh*, *Ptch1*, or *Gli1* on an ABI 7900HT RT-PCR machine. Expression levels were normalized to the geometric mean of three control genes (*Actb, Hprt* and *Ubc*), average normalized Ct values for control and experimental groups determined, and relative expression levels determined by ΔΔCt. The RT-qPCR of each RNA sample was performed in quadruplicate with a minimum of N = 3 biological replicates (biologically distinct samples) per genotype. All primers used for RT-qPCR experiments can be found in *Supplementary file 1*.

## Embryo processing for midface imaging

Embryos were harvested in ice-cold PBS, staged by counting somite number, and fixed o/n at 4 degrees in 4%PFA/PBS. Embryo heads were removed and stained in 0.01 % ethidium bromide in PBS at room temperature for 15 min. Embryos were positioned using glass beads in PBS and imaged on a Leica MZ16 F fluorescence stereomicroscope.

## Image Quantification

For 2D midface width quantification, the infranasal distance was measured using FIJI software by drawing a line between the center of each nasal pit. For quantification of cell death and proliferation assays, a minimum of two sections per embryo were quantified over three biological replicates (biologically distinct samples). Staining with epithelial marker E-cadherin was used for quantification of facial ectoderm while nuclear morphology was used to separate mesenchyme, hindbrain, and forebrain tissue compartments. For quantification, threshold was first set for each image followed by binary watershed separation to obtain accurate nuclei counts. The percentage of TUNEL[+] or pHH3[+] nuclei were compared between *Tctn2* mutant and control samples.

# Additional information

## Competing interests

Jeremy F Reiter: Reviewing editor, eLife. The other author declares that no competing interests exist.

## Funding

| Funder | Grant reference number | Author |
| --- | --- | --- |
| National Institute of Dental and Craniofacial Research | R01DE029454 | Jeremy F Reiter |
| National Institute of Arthritis and Musculoskeletal and Skin Diseases | R01AR054396 | Jeremy F Reiter |
| National Institute of Dental and Craniofacial Research | F30DE024684 | Shaun Abrams |

The funders had no role in study design, data collection and interpretation, or the decision to submit the work for publication.

## Author contributions
Shaun R Abrams, Data curation, Formal analysis, Investigation, Methodology, Validation, Visualization, Writing – original draft; Jeremy F Reiter, Conceptualization, Funding acquisition, Project administration, Writing – review and editing

## Author ORCIDs
Shaun R Abrams (iD) http://orcid.org/0000-0002-1479-9322
Jeremy F Reiter (iD) http://orcid.org/0000-0002-6512-320X

## Ethics
This study was performed in strict accordance with the recommendations in the Guide for the Care and Use of Laboratory Animals of the National Institutes of Health. All mouse protocols were approved by the Institutional Animal Care and Use Committee (IACUC) of the University of California, San Francisco (protocol AN178683).

## Decision letter and Author response
Decision letter https://doi.org/10.7554/eLife.68558.sa1
Author response https://doi.org/10.7554/eLife.68558.sa2

# Additional files

## Supplementary files
• Supplementary file 1. Primer sets used for genotyping and quantitative real-time polymerase chain reaction (RT-qPCR). File contains all sequencing primer pairs used for genotyping of all mouse strains used in this study. In addition, file contains primer sets used for RT-qPCR experiments. Inquiries regarding primers should be sent to Jeremy Reiter.

• Transparent reporting form

## Data availability
All data generated or analysed during this study are included in the manuscript and supporting files.

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
