## [Decision Letter]

**Acceptance summary:**

This paper will be of interest to developmental biologists, cilia researchers and those interested in the etiology of syndromes called ciliopathies. The significance of the study is that it reveals a mechanism by which mutation of proteins that organize the ciliary transition zone can lead to craniofacial patterning defects. Extensive genetic data support the conclusion that loss of these transition zone proteins leads to apoptotic cell death and, ultimately, the resulting smaller structures of the facial midline by reducing Sonic Hedgehog signaling.

**Decision letter after peer review:**

Thank you for submitting your article "Ciliary Hedgehog signaling regulates cell survival to build the facial midline" for consideration by *eLife*. Your article has been reviewed by 2 peer reviewers, and the evaluation has been overseen by a Reviewing Editor and Piali Sengupta as the Senior Editor. The following individual involved in review of your submission has agreed to reveal their identity: Rolf W Stottmann (Reviewer #1).

Essential revisions:

1) The authors observed decreased apoptosis in Tctn2 mutants following reduction of Ptch1 gene dosage but did not see rescue of Gli1 or Shh gene expression in the ventral telencephalon. Based upon these results, the authors propose that it is noncanonical (GLI-independent) PTCH proapoptotic activity that is responsible for the defect, but further evidence is needed to support the claim. Might there be a SHH-PTCH-GLI transcriptional effect that contributes to the rescue but is not detectable by monitoring only Gli1 and Shh expression? Can an attempt be made to directly rule out SHH signaling to GLI effectors? Perhaps reducing Sufu in the Tctn-/- model and testing for rescue of apoptosis vs. target gene expression?

2) If the effect is indeed GLI-independent, some explanation of how primary cilium dysfunction impacts SHH control of PTCH dependence receptor activity is needed, since PTCH-mediated activation of apoptosis likely occurs from the plasma membrane, and not ciliary membrane. Incorporating this into the model in figure 6 would be helpful.

3) The neural crest cells/ mesenchyme make up the vast majority of the cells populating the midface. While these would presumably be at least some of the cells dying there is very little discussion or direct measurement of these cells.

4) Including information about the spatiotemporal expression of the TZ genes in the relevant craniofacial stages would be helpful for interpreting phenotypes.

5) We recommend that all quantifications are shown as in Figure 1 where the data points are included in the graph. Furthermore, p-values should be stated rather than coded with *, **, *** etc.

6) The Discussion reads (line 317), "Our data are consistent with a role of PTCH1 in promoting apoptosis in the neuroectoderm and facial ectoderm which is inhibited by prechordal plate-produced SHH. In the absence of TCTN2, the prechordal plate does not produce SHH, releasing PTCH1 to promote apoptosis in the midline and resulting in midface hypoplasia." However, the data in Figure 2A show that the prechordal plate does continue to express SHH in TCTN2 mutants. Rather, the downstream pathway is affected. This needs to be clarified in the overall model.*Reviewer #1:*

The data presented in this paper support the conclusions drawn. The analyses are standard in the field, well powered and robust. The phenotypes (both gross and molecular) are fairly easy to appreciate. This paper does indeed come to a conclusion about what might be causing the mid facial deficits in the transition zone gene mutant mice: namely, large increased in cell death which are, at least in part, mediated through the Ptc Hh receptor.

Strengths:

Looking at the embryo at the early head fold stages is a nice insight and is a bit earlier than many craniofacial analyses. This represents a nice grounding of this work in fundamentals of embryology.

Significant genetics are done but only briefly discussed (the conditional ablations). These are important as they clearly help guide the authors to the presented model.

Weaknesses:

The neural crest cells/ mesenchyme make up the vast majority of the cells populating the midface. While these would presumably be at least some of the cells dying there is very little discussion or direct measurement of these cells.

The implication of this model is a signal(s) from the ectoderm and the neural ectoderm promoting neural crest cell growth. Further explanation or exploration of this would be a great addition to really understanding the phenotypes.

*Reviewer #2:*

This manuscript reveals a crucial role for primary cilia transition zone components in regulation of proper patterning of the facial midline. Using a genetic knockout strategy, the authors show that loss of transition zone components that are commonly mutated in the ciliopathy Meckel Syndrome (MKS) lead to hypotelorism (narrowing at the facial midline). These mutations were shown to reduce Sonic Hedgehog (SHH) signaling in the prechordal plate, which increased apoptosis in the facial ectoderm and ventral telencephalon. The increased apoptosis and midline defects could be rescued by reducing gene dosage of the SHH receptor and negative signaling regulator Patched (PTCH), suggesting that compromised SHH signaling leads to craniofacial malformation in this genetic context.

Hypotelorism is a common presentation in ciliopathy disorders, but the underlying mechanism has not been clear. The results provided in this study support that pattern disruption likely results from apoptosis of cells in the developing midline neuroectoderm and facial endoderm resulting from reduced SHH-mediated control of dependence receptor activity by PTCH. Intriguingly, target gene analysis in the ventral telencephalon suggests that PTCH signaling to GLI is unlikely to control cell survival during midface development, leading the authors to propose that the pro-apoptotic role of PTCH is the main phenotypic driver in this tissue context. The genetic experiments are clearly presented, and conclusions are supported by the data. The key conclusion of the study – that attenuation of transcription-independent SHH-PTCH signaling is responsible for midline defects – would be strengthened by some supporting mechanistic interrogation (and/or additional genetic interrogation). Monitoring caspase activity suggests that apoptosis is impacted, but the conclusion that there is unlikely to be a GLI-dependent transcriptional component contributing to patterning alteration is not appropriately supported.

---

## [Author Response]

Essential revisions:1) The authors observed decreased apoptosis in Tctn2 mutants following reduction of Ptch1 gene dosage but did not see rescue of Gli1 or Shh gene expression in the ventral telencephalon. Based upon these results, the authors propose that it is noncanonical (GLI-independent) PTCH proapoptotic activity that is responsible for the defect, but further evidence is needed to support the claim. Might there be a SHH-PTCH-GLI transcriptional effect that contributes to the rescue but is not detectable by monitoring only Gli1 and Shh expression? Can an attempt be made to directly rule out SHH signaling to GLI effectors? Perhaps reducing Sufu in the Tctn-/- model and testing for rescue of apoptosis vs. target gene expression?

As the reviewers points out, we include in the Discussion comment on the possibility that SHH relieves the proapoptotic activity of PTCH to support growth of the facial midline. However, other possibilities, including the idea that transcriptional effects independent of general HH targets such as Gli1, are not disproven. Therefore, we include discussion of these other possibilities in the revised Discussion.

As suggested by the reviewers, we attempted to rule out the involvement of SHH signaling to GLI effectors in the survival of midline cells. Unfortunately, generating *Tctn2 Sufu* double mutant mice was not feasible in the time allotted for the revision process. We reasoned that SAG, a Smoothened agonist, should activate GLI transcriptional effects but not antagonize the biochemical functions of PTCH, whereas SHH should activate GLI transcriptional effects and antagonize the biochemical functions of PTCH. Thus, an alternative test of whether SHH signaling to GLI effectors is critical for the survival of midline cells is to assess whether both SAG and SHH have similar or distinct effects on midline cell survival. If GLI activation is sufficient to promote midline cell survival, both SHH and SAG should increase cell survival in wild-type embryos but not in *Tctn2* mutant embryos. If, instead, biochemical inhibition of PTCH is critical for midline cell survival, SHH (an inhibitor of PTCH) should increase cell survival even in *Tctn2* mutant embryos, whereas SAG (which acts downstream of PTCH) should not. Rescue of *Tctn2*^-/-^ cell viability by SHH but not SAG would further suggest that unopposed PTCH1 activity inhibits cell viability in the absence of transition zone function.

To attempt to test these hypotheses, we employed a Whole Embryo Culture (WEC) system to culture of E8.5 mouse embryos ex utero in WEC-grade rat serum. We successfully cultured E8.5 embryos for 24 hours. Cultured embryos developed similarly to in utero embryos, as determined by turning, closure of the neural tube and looping of the heart tube.

Despite the success establishing the WEC system, it was unclear whether SAG or SHH added to the medium will access HH responsive cells within the embryo. Unfortunately, we were unable to detect robust induction of *Ptch1* and *Gli1* over a range of SHH and SAG concentrations. Thus, as directed by the reviewers, we have made an attempt to rule out SHH signaling to GLI effectors, but have not definitively proven that GLI effectors are uninvolved in inhibiting midline apoptosis. Therefore, we have revised the Discussion to acknowledge the possibility that SHH signaling to GLI effectors may contribute to midline growth in the craniofacial complex. Some of these changes are incorporated in the penultimate paragraph of the Discussion.

2) If the effect is indeed GLI-independent, some explanation of how primary cilium dysfunction impacts SHH control of PTCH dependence receptor activity is needed, since PTCH-mediated activation of apoptosis likely occurs from the plasma membrane, and not ciliary membrane. Incorporating this into the model in figure 6 would be helpful.

We concur with the reviewers that PTCH-mediated activation of apoptosis likely occurs at the plasma membrane. We propose that primary cilium dysfunction impacts PTCH-mediated apoptosis by controlling expression of SHH in the ventral telencephalon. Although beyond the scope of this manuscript focused on craniofacial development, we speculate that in other tissues, such as the developing spinal cord, reduced SHH expression caused by ciliary dysfunction will similarly cause increased apoptosis. In yet other tissues, such as the limb bud (where ciliary dysfunction does not alter *Shh* expression), we predict that ciliary dysfunction will not cause increased apoptosis. Thus, ciliary dependence of *Shh* expression may be a determinant of tissues in which ciliary dysfunction result in increased apoptosis. We now include a brief description of these possibilities in the new seventh paragraph of the Discussion.

As suggested by the reviewers, we also revised model in figure 6 to highlight that PTCH-mediated activation of apoptosis could initiate from the plasma membrane. In addition, we now include explicit description of this possibility in the penultimate paragraph of the Results section.

3) The neural crest cells/ mesenchyme make up the vast majority of the cells populating the midface. While these would presumably be at least some of the cells dying there is very little discussion or direct measurement of these cells.

We apologize for our lack of clarity. We measured cell death (via TUNEL, cleaved Caspase-3 staining, and cleaved Caspase-9 staining) and proliferation (via phospho-Histone H3 staining) in the midface. Further quantification has been added in review. and show that there is a statistically significant increase in cell death in the mesenchyme (as well as the brain and midline ectoderm) at E9.5. The increase in cell death in the mesenchyme is now also discussed in the revised text (including in the penultimate paragraph of the Results section) and Figure 4.

4) Including information about the spatiotemporal expression of the TZ genes in the relevant craniofacial stages would be helpful for interpreting phenotypes.

Prior spatiotemporal expression of transition zone genes has revealed that these genes are widely expressed throughout the stages analyzed here. We have included text that references our and others’ previous studies that detail expression pattern of transition zone genes at relevant stages for our current study. The revised text is included in the second paragraph of page 7 in the Results section.

5) We recommend that all quantifications are shown as in Figure 1 where the data points are included in the graph. Furthermore, p-values should be stated rather than coded with *, **, *** etc.

As recommended by the reviewers, we have changed all quantifications to explicitly show all data points, as previously done for Figure 1. In addition, we have now included stated p-values in all figures.

6) The Discussion reads (line 317), "Our data are consistent with a role of PTCH1 in promoting apoptosis in the neuroectoderm and facial ectoderm which is inhibited by prechordal plate-produced SHH. In the absence of TCTN2, the prechordal plate does not produce SHH, releasing PTCH1 to promote apoptosis in the midline and resulting in midface hypoplasia." However, the data in Figure 2A show that the prechordal plate does continue to express SHH in TCTN2 mutants. Rather, the downstream pathway is affected. This needs to be clarified in the overall model.

Thank you for pointing out this error. We have revised the text (in the second paragraph of page 14) to state, “In the absence of TCTN2, the ventral telencephalon does not produce SHH, releasing PTCH1 to promote apoptosis in the midline and resulting in midface hypoplasia.”